# Effects of Glucose and Corn Syrup on the Physical Characteristics and Whipping Properties of Vegetable-Fat Based Whipped Creams

**DOI:** 10.3390/foods11091195

**Published:** 2022-04-20

**Authors:** Yongchao Zeng, Di Zeng, Tongxun Liu, Yongjian Cai, Yonghao Li, Mouming Zhao, Qiangzhong Zhao

**Affiliations:** 1School of Food Science and Engineering, South China University of Technology, Guangzhou 510640, China; zengyc0929@163.com (Y.Z.); dickzengen@163.com (D.Z.); txliu@scut.edu.cn (T.L.); cai2097@163.com (Y.C.); lllyonghao@163.com (Y.L.); femmzhao@scut.edu.cn (M.Z.); 2Research Institute for Food Nutrition and Human Health, Guangzhou 510640, China

**Keywords:** glucose, corn syrup, emulsion characteristics, fat coalescence, whipped cream properties

## Abstract

The aim of this work is to evaluate the effects of glucose and corn syrup on the physical characteristics and whipping properties of whipped creams. The interfacial protein concentration and apparent viscosity of emulsions increased with an increasing sugar concentration. In whipped creams, a shorter optimum whipping time (*t_op_*), higher fat coalescence degree, higher firmness and higher stability were detected as sugar concentration increased. The partial coalescence degree, overrun and firmness of whipped cream with 30 wt% glucose reached 76.49%, 306% and 3.82 N, respectively, significantly (*p* < 0.05) higher than those (67.15%, 235% and 3.19 N) with 30 wt% corn syrup. Compared with glucose at the same sugar concentration, higher interfacial protein concentration and less-shaped aggregates and coalescences were observed for the emulsions upon the addition of corn syrup, which caused a lower degree of fat coalescence and a lower firmness of whipped cream. The differences could be explained by the presence of maltodextrin (MDX) in corn syrup, which protects absorbed protein throughout freezing and retards the formation of a continuous network during whipping. As a result, the addition of sugars could well improve stability of emulsion, firmness and foam stability of whipped cream efficiently. With a 25–30 wt% sugar addition, even if there was a lower partial coalescence degree and firmness compared with glucose, whipped cream with corn syrup exhibited relatively good stability. These results suggest that MDX improves the stability of emulsion and, thus, has a potential use in low-sugar whipped cream.

## 1. Introduction

Whipped cream, a popular dairy product, is broadly used in desserts, cakes and other products after aeration [1]. A whipped cream emulsion, containing water, sugars, fat, protein, stabilizers and emulsifiers, should keep relatively stable during freezing [2]. During whipping, a stable fat network composed of partially coalesced globules and interfacial protein layers is established. Fat influences crystallization behavior and partial coalescence of fat, which can consequently affect the whipping properties [3]. Proteins can reduce the interfacial tension and form a barrier between the oil and water phases [4], thereby increasing the stability of whipped cream emulsion. Emulsifiers can absorb the interface, thereby lowering the interfacial tension, or alter the properties of fat crystallization [5,6]. Stabilizers are usually used to modify the viscosity of the aqueous phase [7].

Vegetable-fat-based whipped cream formulated from glucose and corn syrup is growing in popularity due to its lower cost and higher stability in comparison with butter-based whipped cream. However, the content of sugar reaches 20–30 wt% in markets. High levels of sugar do not match the need for nutrition and health. Therefore, it is very necessary to decrease the content of sugar and develop new, healthy products of whipped cream. In order to prepare low-sugar whipped cream, it is essential to explore the interactions between sugar and the main ingredients in whipped cream, and to understand the role of sugar in whipped cream during the whipping process.

Sugar is a common ingredient of aerated foods, which provides sweetness, texture and structure in whipped cream [8]. To understand how sugar affects the stability and quality of whipped cream, it is useful to look at simple simulation systems of ice cream and cake. The effect of sugar on their properties is already extensively reported in the literature.

Sugars not only fulfill the taste and flavor, but also are involved in a number of functions, such as binding water, increasing boiling temperature, lowering the freezing temperature of aqueous solutions, increasing viscosity and altering the behavior of proteins, starches, and hydrocolloids [9].

In an aerated system containing 2–6 wt% egg albumen, reduction in sugar content accelerates the foam destabilization processes, whereas it enhances the overrun of foam due to the decrease in drainage rate and viscosity of the liquid continuous phase [10]. In an oil-in-water frozen emulsion, fat globules force closer together with the formation of ice crystals in the serum phase. Then, the free water present may be insufficient to fully hydrate the emulsifier or protein adsorbed to the droplet surfaces, thereby facilitating the droplet–droplet interactions. Finally, ice crystals possibly form during freezing, which may pierce the interfacial layer between the oil droplets, thus making them more liable to coalescence once they are thawed [11]. Consequently, the presence of sugar may influence the size and morphology of ice crystals because it affects crystal growth during freezing. Besides, sugar also influences the functional properties of proteins at the oil-water interface, such as adsorption and gelation.

In ice cream, sugar helps depress the freezing point, which reduces the formation of large ice crystals that contribute to a gritty mouthfeel. If all water is frozen, the texture of ice cream will be too hard to be accepted by consumers. MDX is a hydrolysis product of starches with a different dextrose equivalent (DE), which is used in food emulsions as a stabilizer. A lower DE corresponds to a higher degree of polymerization [12]. MDX with low DE values can protect emulsions against freezing destabilization, improve the sensory attributes of ice cream and produce strong stickiness [9,13].

In cake products, sugar acts as a tenderizer, which can weaken or prevent the development of a gluten network during cake batter mixing by competing with gluten proteins for water [14]. The same phenomenon can be observed in aerated confectionery [10,13]. Sugar also influences the color of the cake due to the Maillard reaction [15].

In summary, sugar can influence properties of emulsions; therefore, it probably affects the whipping properties and quality of whipped cream. However, considerable research has focused on sugar in relation to the physical characteristics and freeze-thaw stability of ice cream products or simple simulation systems. There are still uncertainties in the understanding about the effect of sugar on the complex emulsion-based foam system. Furthermore, in recent years, more and more people are suffering from obesity and type 2 diabetes, which is potentially related to the high content of sugars in diets [9,16]. Many enterprises are actively developing new whipped cream products desirable to consumers, such as low-sugar products.

Therefore, the objective of this work was to elucidate the correlation between the addition of sugars (type and concentration) and the properties of whipped cream. The commonest sugars (glucose and corn syrup) with greatly different structures were selected to prepare whipped creams. This study examined the physical characteristics of an emulsion before whipping and the whipping properties of whipped cream, and then explored the correlation between emulsion characteristics and whipped cream quality. This work will not only provide a better understanding of the mechanism of sugars on the whipping properties, but also contribute to the development of healthier whipped cream with low sugar.

## 2. Materials and Methods

### 2.1. Materials and Reagents

Hydrogenated palm kernel oils (BL-41; HPKO; C12:0, 51.1 g/100 g, C14:0, 15.3 g/100 g, C16:0, 8.9 g/100 g, C18:0, 17.3 g/100 g) were donated by Cargill Grain & Oilseeds Ltd. (Dongguan, China). Compound stabilizers 8022, composed of polyglyceryl ester of fatty acid, sodium stearyl lactate, sucrose ester S1170, xanthan gum, carrageenan, guar gum, sodium dihydrogen phosphate and dibasic sodium phosphate, were supplied by Guangdong Wenbang Biotechnology Co. (Zhaoqing, China). Sodium caseinate (≥95 wt%) was obtained from New Zealand Milk Products (NZMP, Fonterra Co-operative Group Limited, Auckland, New Zealand). Maltodextrin (soluble solids content > 94%; DE, 19.1) and glucose (soluble solids content > 90%) were purchased from Zhucheng DongXiao Biotechnology Co. (Zhucheng, China). Corn syrup (soluble solids contents > 75%; DE, 42; MDX, 35%; Zhaoqing Huanfa Biotechnology Co., Zhaoqing, China) was purchased from Ligao Foods Ltd. (Guangzhou, China).

### 2.2. Emulsion Preparation

Hydrogenated palm kernel oils (16 wt%) were melted at 65 °C; subsequently, sodium caseinate (0.5 wt%) and compound stabilizers (1.5 wt%) were added to obtain the oil phase. The aqueous phase contained glucose or corn syrup at different concentrations (10, 15, 20, 25 or 30 wt%), and deionized water. In addition, 30 wt% sugar concentrations with different ratios of MDX and glucose (0:10, 1:19, 2:8, 3:7, 4:6) were prepared to verify the effect of MDX on the emulsion stability. The two phases were blended together and stirred at 750 rpm and 65 °C for 30 min with a digital stirrer (RW20, IKA Co., Staufen, Germany). Then, the mixtures were immediately homogenized twice by a homogenizer (APV-1000, Kolding, Denmark) at a pressure of 500 bar to obtain emulsions and, finally, were frozen at −18 °C for 24 h to obtain frozen emulsions.

### 2.3. Emulsion Properties

#### 2.3.1. Microstructure Observation

The emulsions were diluted 10 times with deionized water, placed on a glass slide and covered with a cover slip for microstructure observation, according to the method described by Zhao et al. [6] with slight modifications (objective lens 40× and eyepiece 10×). Microstructure images were obtained via an Olympus CX31 light microscope (Olympus, Tokyo, Japan) equipped with an MShot MD130 digital camera (Guangzhou Mingmei Technology Co., Ltd., Guangzhou, China), which were processed by the software Micro-Shot Basic (version 1.0).

#### 2.3.2. Particle Size Distribution

Particle size distribution of fat globules was measured by an integrated laser light scattering instrument (Mastersizer 2000, Malvern Instruments Co. Ltd., Worcestershire, UK) following the method described by Long et al. [17] with modifications. Relative refractive index and absorption were set as 1.414 and 0.001, respectively. The emulsion was diluted approximately 1000 times with distilled water at ambient temperatures.

#### 2.3.3. Interfacial Protein Concentration

Interfacial protein concentration was measured according to the method described by Long et al. [17] with slight modifications. Emulsions were centrifuged at 10,000× *g* for 30 min at 25 °C via a Beckman L8-M ultracentrifuge (Beckman, Fullerton, CA, USA). The subnatants were carefully removed by a syringe and were separated by membrane filtration. The protein concentration was measured by the micro-Kjeldahl procedure, using 6.25 as the protein conversion factor. The interfacial protein concentration (C, mg/m^2^) and specific surface area (SSA, m^2^/g) were calculated by Equations (1) and (2), respectively:(1)$$C=\frac{M_{t}-M_{s}}{F_{m}\times \mathrm{SSA}}\times 1000$$
(2)$$\mathrm{SSA}=\frac{6}{{\rho \times d}_{3,2}}$$
where *M_t_* (g) and *M_s_* (g) were the mass of the protein in the emulsion and subnatant, respectively, *F_m_* (g) was the mass of the cream layer, *ρ* (g/mL) was the density of the emulsion and *d*_3,2_ was the surface area mean diameter calculated by the Mastersizer 2000.

#### 2.3.4. Apparent Viscosity

Apparent viscosity of the emulsions was detected by a Mars III rheometer (Thermo Haake Co. Ltd., Karlsruhe, Germany) with a P35 TiL polished sensor, according to the method described by Zhao et al. [6] with slight modifications. The measurements were performed at a gap distance of 1 mm at 25 ± 1 °C, and the shear rate was linearly increased from 0.1 to 100 s^−1^ for 300 s. The data were analyzed by the Rheowin Data Manager software version 4.30 (Thermo Haake Co. Ltd., Karlsruhe, Germany).

### 2.4. Whipped Cream Preparation

The emulsions (800 ± 10 g) were thawed to 0–4 °C and were whipped with a kitchen mixer (KM800, Kenwood Ltd., Woking, UK) at speed setting 5 (approximately 160 rpm). All emulsions were whipped until a defined observable end point, at which a cream was obtained and broke away from the wires and the bowl [18]. Whipping experiments were repeated at least three times for each cream. The water distribution, overrun, firmness, serum loss and microstructure were determined.

### 2.5. Whipped Cream Properties

#### 2.5.1. Low-Field Pulsed NMR (LF-NMR)

In order to evaluate the effect of sugars on water distribution in the whipped cream, NMR relaxation measurements were carried out through LF-NMR, using the methods described by Zhao et al. [19] with some modifications. Approximately 5 g of each whipped cream sample was transferred to a 15 mL serum bottle at 4 °C for 30 min, and then was placed immediately inside a 40 mm cylindrical probe of a Niumag Low-Field Pulsed NMR analyzer (NMI 20-040H-I, Niumag, Suzhou, China) with a magnetic field strength of 0.5 T. The analyzer was operated at 4 °C and a resonance frequency of 20 MHz. The transverse relaxation time T_2_ was measured using the Carr–Purcell–Meiboom–Gill sequence with 6 scans, 12,000 echoes, 4 s between scans and 480 μs between pulses of 90° and 180°. The data were fitted to a multiexponential curve via the Niumag MultiExp Inv Analysis software 4.09.

#### 2.5.2. Partial Coalescence of Fat

Fat particle size distribution determined by a laser diffraction technique is a reliable and fast method to quantify the partial coalescence in whipped cream. Each whipping batch was collected immediately at the *t_op_*; then, the particle size distribution was measured at ambient temperature. The cumulative percentage of the particles at 3.0 μm were recorded. This value was taken as a measure of the formation of a second peak of fat globule aggregates [20,21,22]. The correlative indexes were consistent with the indexes used in Section 2.3.2.

#### 2.5.3. The Optimal Whipping Time (*t_op_*)

The optimal whipping time (*t_op_*) was defined as the time at which a cream was obtained and broke away from the wires and the bowl [18].

#### 2.5.4. Overrun

Overrun was measured according to the method described by Zhao et al. [6], which was related to the mass of this volume and the density of the cream before whipping. It was determined by Equation (3).
(3)$$\mathrm{Overrun}\left(\%\right)=\frac{M_{1}-M_{2}}{M_{2}}\times 100$$
where *M*_1_ (g) and *M*_2_ (g) represented the mass of the cream before whipping and after whipping, respectively.

#### 2.5.5. Firmness

The firmness of whipped cream was calculated with a TA-XT Texture Analyzer (Stable Micro Systems, Godalming, UK) according to the method described by Liu et al. [23]. Measurements in compression mode and an A/BE probe (35 mm diameter) were selected. Puncture tests were conducted at a rate of 1 mm⋅s^−1^ over a distance of 25 mm in the sample. The trigger value for the start of the measurement was set to 0.20 N. The force (N) required to reach this depth was defined as the firmness of the whipped cream.

#### 2.5.6. Microstructure Observation

The air bubble distribution of whipped cream at the *t_op_* was observed via a polarized light microscope (PLM) (Olympus, BX-41 P, Tokyo, Japan) attached to a microscopy imaging system (Olympus, DP22, Tokyo, Japan). The cream was placed on the glass slide and covered with the cover slip for observation, according to the method described by Zeng et al. [2]. Air bubble size distribution was analyzed by using a manual analysis procedure of ImageJ 1.53k software (USA) [24].

#### 2.5.7. Serum Loss

Whipped cream (45 ± 5 g) was transferred to a funnel and placed on top of an Erlenmeyer. Then the funnel was placed in the incubator at 25 °C for 12 h. A tared beaker was used to collect serum and syneresis was calculated using Equation (4) to evaluate the foam stability [18]:(4)$$\mathrm{Serum}\;\mathrm{loss}\left(\%\right)=\frac{m_{\mathrm{serum}}}{m_{\mathrm{whipping}}}\times 100$$
where *m_serum_* (g) represented the mass of cream poured through the funnel, and *m_whipping_* (g) represented the initial mass of whipped cream transferred to the funnel.

#### 2.5.8. Cream Performance after Storage

Whipped cream was piled into hill shapes and was placed at 25 °C for 6 h. The degree of roughness of the section surface inside the cream was observed and photographed by a digital camera.

### 2.6. Statistical Analysis

All the tests were performed and reported as mean ± standard deviation. An analysis of variance (ANOVA) was performed using the SPSS 21.0 statistical analysis program. All of the experiments were performed in triplicate, and the differences were considered statically significant at a confidence level of 95%.

## 3. Results and Discussion

### 3.1. Emulsion Properties

#### 3.1.1. Particle Size Distribution and Microstructure of Emulsion

Figure 1A,B show particle distributions of the emulsions before whipping with 10, 15, 20, 25 and 30 wt% of glucose and corn syrup added. The particle size distribution of all samples showed a principal peak in the lower size range with a shoulder or tail peak in the higher size range. With the increase in sugar content, the principal peak became higher, and the shoulder or tail peak narrowed down and turned into a lower size range. As sugar increased from 10 to 30 wt%, the average particle size (*d*_3,2_) of the emulsions with glucose decreased from 0.195 to 0.159 μm, whereas it decreased from 0.184 to 0.153 μm for the emulsions with corn syrup. In addition, the effect of sugars on the microstructure of emulsions before whipping was observed (Figure 1C,D). As sugar concentration increased, the image revealed fewer large aggregates and smaller fat droplets. However, at the same sugar concentration, larger and more shaped aggregates were observed for the emulsions with glucose.

The *d*_3,2_ and droplet aggregate degree are indicators in emulsions that affect emulsion stability. The effects of sugar level on the droplet size and the number of large aggregates could be attributed to the following two reasons. On the one hand, the increase in soluble solids content upon the addition of sugar led to the higher proportion of oil in the unit volume of the emulsion, which caused the higher crushing effect during the homogenization process, thereby leading to the smaller average particle size of the emulsion. On the other hand, the presence of sugar may reduce the frequency of collisions between fat droplets by acting as a spacing matrix [25,26,27]. Consequently, as the sugar concentration increased, the collision of fat globules slowed down, thereby decreasing the degree of droplet aggregate and coalescence.

However, at the same sugar concentration, a fewer number of irregularly shaped aggregates was observed for the emulsions with corn syrup, indicating that corn syrup could be more effective in decreasing fat droplet flocculation than glucose. This might be due to the existence of MDX in corn syrup. To verify this hypothesis, the microstructure and the particle distributions of emulsions with different ratios between MDX and glucose were presented in Appendix A and analyzed. As the ratio between MDX and glucose increased from 0:10 to 4:6, the micrograph became more uniform with fewer large fat droplets, and the *d*_3,2_ of the emulsion decreased from 0.152 to 0.142 μm. These results indicated that MDX could enhance the stability of an emulsion and protect fat droplets against flocculation, aggregation or coalescence, which was similar to the previous works [28,29]. MDX occupies a larger volume in an emulsion compared to glucose, thus retarding the collision of droplets forming aggregates and coalescence during the freezing and thawing process. Hence, the emulsion with corn syrup exhibited smaller droplet sizes and fewer numbers of aggregates, which was mainly due to the MDX.

In addition, at 30 wt% sugar concentration, the microstructure of emulsions with the addition of MDX and glucose at the ratio of 3:7 and 4:6, respectively, was relatively close to that of emulsions with corn syrup, of which the content of MDX was 35 wt%.

#### 3.1.2. Interfacial Protein Concentration of Emulsion

Determining the extent of protein adsorption is a straightforward method that obtains information about adsorbed layers [6]. The effect of different sugars on the interfacial protein concentration before whipping is shown in Figure 2. As sugar concentration increased from 10 to 30 wt%, the interfacial protein concentration increased from 0.34 to 0.79 mg/m^2^ in emulsions with glucose and from 0.50 to 0.77 mg/m^2^ in emulsions with corn syrup.

Interfacial protein concentration depends on specific surface area (SSA) and total interfacial proteins load together [2]. Upon the addition of sugar, the droplet size was decreased (Figure 1) and the density of the emulsion was increased, which together caused the SSA to increase slightly from 29.80 to 34.20 m^2^/g and from 31.40 to 35.79 m^2^/g for the emulsions with glucose and corn syrup, respectively. The interfacial protein content increased significantly from 10.12 to 27.11 mg/g as glucose concentration increased from 10 to 30 wt%, whereas that of corn syrup increased from 15.73 to 26.11 mg/g.

As sugar concentration increased, the increase of interfacial protein concentration could be attributed to the decrease in the size of ice crystals during freezing, which reduced the compressive force on the surface of fat droplets. The large ice crystals formed at a lower concentration of sugar during freezing in the aqueous phase could pierce the interfacial protein film, resulting in the desorption of proteins from the interface [30]. Therefore, as sugar concentration increased, the formation of smaller-sized ice crystals could not destroy the protein layer, causing higher interfacial protein content.

It was worth noting that the interfacial protein concentration was lower for the emulsions with glucose than that with corn syrup at a lower sugar concentration (10–15 wt%). This was mainly due to the MDX acting as a macromolecular in corn syrup, which could restrict water molecules from forming larger-size ice crystals [31]. The fewer larger-size ice crystals during freezing was more effective at protecting the adsorbed protein at the interface. Hence, there was a higher interfacial concentration in emulsions with corn syrup addition.

#### 3.1.3. Apparent Viscosity of Emulsion

The apparent viscosity vs. shear rate for emulsions with glucose and corn syrup at different concentrations are presented in Figure 3A,B. All the emulsions exhibited shear-thinning behavior. As sugar concentration increased from 10 to 30 wt%, the apparent viscosity of emulsions with glucose and corn syrup increased from 0.35 to 0.81 Pa s and from 0.14 to 0.60 Pa s at a shear rate of 5.8 s^−1^, respectively, and at shear rate of 100 s^−1^, it increased from 0.05 to 0.15 Pa s and from 0.04 to 0.10 Pa s for the emulsions with glucose and corn syrup, respectively.

It has been reported that smaller droplet size and higher soluble solids content increases the viscosity of an emulsion system [32]. At a shear rate of 5–40 s^−1^, the addition of sugar led to a decrease in average droplet size (Figure 1A,B) and an increase in interparticle resistance force to flow, thereby resulting in the increase of viscosity. At a shear rate of 40–100 s^−1^, as sugar concentration increased, the decreasing rate of apparent viscosity increased gradually, indicating that the internal structures of the emulsions were slowly destroyed, which meant that fat globules had a significant effect on viscosity.

Furthermore, at a fixed shear rate, the apparent viscosity of the emulsions with glucose groups was higher than that of those with corn syrup at the same sugar concentration. At the same concentration, there were more fat coalescence and aggregates in the emulsions with glucose than those with corn syrup (Figure 1C,D), and there were looser structures in emulsions with corn syrup than those of glucose, which could cause higher viscosity. Moreover, higher serum protein concentration in glucose samples can increase viscosity. A similar result was also reported by Pearce et al. [33].

### 3.2. Whipped Cream Properties

#### 3.2.1. Relaxation Time Analysis

A fitted T_2_ distribution is used to assess the mobility and proportion of different fractions of water molecules in the gel system. Figure 4A,B present the T_2_ relaxation time distributions for creams with glucose and corn syrup, respectively. Three peaks were observed for creams with glucose, whereas four peaks were noted for creams with corn syrup. NMR transverse relaxation of water (T_2_) can be separated into the relaxation decay of three exponential populations, representing three water compartments: bound water (T_2b_ 0–10 ms), immobilized water (T_21_ 10–100 ms), and free water (T_22_ 100–1000 ms) [34].

The T_2b_ of creams with glucose and corn syrup decreased with the increase in sugar concentration, and the proportion of T_2b_ peaks increased from 0.97 to 3.46% and from 2.80 to 3.52% for creams with glucose and corn syrup, respectively, indicating that bound water had a lower water mobility and was more closely associated with the protein. It has been reported that the water-protein interactions were less effective due to lower water mobility in the presence of sugar, which might enhance hydration and reduce the relaxation time of T_2b_ [35]. At the same soluble solids content, the bound water of creams with corn syrup was more closely associated with protein due to the higher water-binding ability of MDX, which could affect the interfacial adsorption of protein.

As sugar concentration increased from 10 to 30 wt%, the proportion of T_21_ peaks of creams with corn syrup increased from 3.30 to 11.89%, whereas it showed little difference for the creams with glucose. Additionally, the T_22_ peak relaxation time of creams with glucose and corn syrup shifted from 943.79 to 505.26 ms and from 821.43 to 357.08 ms, respectively. As sugar concentration increased, the higher soluble solids content led to less mobile water and shorter relaxation time. However, at the same sugar concentration, the proportions of T_21_ and T_22_ of creams with corn syrup were more than those with glucose, indicating that water was more closely associated with corn syrup than glucose. MDX is widely used in the encapsulation of food components. Longer chains of MDX could connect aggregates to form a continuous network, in which water is trapped as immobilized water [36]. Therefore, the water mobility was restricted and more water was left in the network.

#### 3.2.2. Partial Coalescence of Whipped Cream

Figure 5 indicates the extent of partial coalescence of whipped creams with glucose and corn syrup. As sugar concentration increased from 10 to 30 wt%, the extent of partial coalescence of creams with glucose increased from 57.36 to 76.49%, whereas that of corn syrup increased from 22.71 to 67.15%.

During whipping, adjacent fat globules approach each other and coalesce partially under mechanical shear. Therefore, the collision frequency and partial coalescence efficiency of fat globules are important factors for the speed and extent of partial coalescence. As sugar concentration increased, the increase in the number of fat globules in unit volume led to the increase in collision frequency of fat globules, which might accelerate partial coalescence of fat globules. Moreover, the remaining small crystals were then no longer kept in a continuous network; thus, they could move freely to the energetically favored oil-water interface upon heating during the whipping process [18]. As the sugar concentration increased, the rates of heating increased during whipping; thus, more fat crystals were moved to the interface at the same whipping time, thereby leading to the higher speed and extent of partial coalescence.

In addition, at the same sugar concentration, the extent of partial coalescence was higher for creams with glucose than those with corn syrup. For the sample with corn syrup, MDX with longer chains could form a continuous network to decrease the frequency of fat droplet collision during the whipping process, causing a lower fat coalescence degree of whipped creams. Higher interfacial protein concentration would retard partial coalescence efficiency, thus leading to a lower degree of partial coalescence as well.

#### 3.2.3. Interfacial Protein Concentration of Whipped Cream

The effect of sugars on the interfacial protein concentration of whipped creams at the *t_op_* is presented in Figure 6. As sugar concentration increased, the interfacial protein concentration of whipped creams with glucose and corn syrup increased from 2.72 to 7.23 mg/m^2^ and from 0.86 to 3.18 mg/m^2^, respectively. In addition, interfacial protein content of creams with glucose increased from 10.38 to 21.39 mg/g, whereas that of creams with corn syrup increased from 15.88 to 26.02 mg/g. SSA of creams with glucose decreased from 3.82 to 2.96 m^2^/g, whereas that of creams with corn syrup decreased from 18.56 to 8.19 m^2^/g (Appendix A). Consequently, the increase of interfacial protein concentration could be attributed to the faster decrease in SSA and increase in the total surface protein load. At the same sugar concentration, the interfacial protein concentration was higher for creams with glucose than those of corn syrup, which was due to a lower SSA caused by a higher extent of fat coalescence.

#### 3.2.4. The Optimum Whipping Time of Whipped Cream

The whipping properties of whipped creams with different sugars at different concentrations are presented in Table 1. As sugar concentration increased from 10 to 30 wt%, the *t_op_* of creams with glucose decreased from 980 to 390 s, whereas that of creams with corn syrup decreased from 1033 to 510 s. A shorter *t_op_* indicates an increased rate of partial coalescence at the air interface in cream [37,38]. During whipping, as sugar concentration increased, emulsions with a higher rate of partial coalescence needed less *t_op_* to form an aerated structure to stabilize eventual products. Moreover, at the same concentration of sugar, a higher partial coalescence rate of glucose samples could lead to a shorter *t_op_* compared to corn syrup.

#### 3.2.5. Overrun of Whipped Cream

At the *t_op_*, the overrun of whipped creams with corn syrup decreased significantly from 376 to 235% as sugar concentration increased from 10 to 30 wt% (Table 1), whereas in glucose samples, the overrun firstly increased from 263 to 335% as sugar concentration increased from 10 wt% to 20 wt% and then decreased to 306% as sugar concentration increased to 30 wt%.

In corn syrup samples, as sugar concentration increased, a higher rate of fat coalescence reduced the *t_op_*, and thus, less time was necessary for air to be incorporated. Reduction of serum protein concentration could also decrease the overrun as corn syrup concentration increased [6]. During the first stage of whipping, a large amount of air was whipped into large bubbles, which were then broken up into smaller bubbles, including bubble break-up and fat coalescence [39]. In glucose samples, as sugar concentration increased from 10 to 20 wt%, faster partial coalescence led to less time of the first stage and the volume of beaten air decreased, but the stronger partial coalescence structure can prevent the bubble break-up. Therefore, at a lower sugar concentration (10–15 wt%), even with a longer *t_op_*, a weaker fat coalescence structure cannot entrap more air in the eventual whipped cream, thus causing lower overrun. As sugar concentration increased to 20%, more air was well entrapped in the stronger fat coalescence structure [39]. Whereas, as sugar concentration increased further to 30%, the shorter *t_op_* and less serum protein concentration caused lower overrun.

#### 3.2.6. Firmness of Whipped Cream

As showed in Table 1, the firmness of creams with glucose increased greatly from 1.25 to 3.82 N as sugar concentration increased from 10 to 30 wt%, whereas it increased from 1.23 to 3.19 N for the creams with corn syrup. Firmness is related to the soluble solids content, interfacial protein concentration, overrun and extent of partial coalescence of whipped cream [18,40].

As sugar concentration increased, the soluble solids content, interface protein concentration and fat coalescence increased; therefore, a more effective structure of whipped cream was formed to stabilize the aerated system, thereby leading to higher firmness [41].

At a lower concentration (10–20 wt%), similar firmness was observed for creams with glucose and corn syrup. Even if there was a relatively higher degree of fat coalescence and interfacial protein concentration in glucose samples, the MDX in corn syrup can help form a stronger structure, thus enhancing the firmness of whipped cream. As sugar concentration increased from 20 to 30%, the fat coalescence and interfacial proteins increased significantly, which played a main role in firmness. Hence, the firmness of glucose samples with a higher fat coalescence degree and interfacial protein concentration was higher than those of corn syrup.

#### 3.2.7. Microstructure of Whipped Cream

The PLM images and the corresponding air bubble frequency distributions of whipped creams with glucose and corn syrup are shown in Figure 7. The microstructure of whipped cream was monitored with PLM to shed light on the network of clumped fat globules at different concentrations. It could be clearly observed that the size of air bubbles became more uniform as sugar concentration increased. A long whipping time as sugar concentration increased caused small air bubbles that incorporated into a larger one; thus, there were more large bubbles in samples at a lower sugar concentration. Whipped cream with a lower overrun and higher extent of fat coalescence could stabilize air bubbles by a stronger network at a higher concentration (20–30 wt%). Furthermore, at 25 and 30 wt% of sugar concentration, the average bubble size of whipped creams with glucose was relatively bigger and more uniformly distributed than that of corn syrup at the same sugar concentration, which was mainly due to higher overrun, firmness, interfacial protein concentration and stronger fat networks of creams. 

#### 3.2.8. Serum Loss and Performance of Whipped Cream after Storage

The appearance of the cross-section of creams after 6 h is shown in Figure 8A,B, and serum loss of creams is presented in Table 1. The cross-section and serum loss could reflect the stability of whipped cream [2]. As sugar concentration increased from 10 to 30 wt%, a smoother cross-section, fewer irregular bubbles and lower serum loss were observed. This might be explained by the increased foam stability of whipped cream due to the higher soluble solids content, thicker protein layers and stronger fat crystals network.

At a lower sugar concentration (10–20 wt%), the samples with glucose showed higher stabilities due to a high interfacial protein concentration and degree of fat coalescence compared to that of corn syrup. However, at a higher sugar concentration (25–30 wt%), even the interfacial protein concentration and fat coalescence degree of whipped cream with corn syrup were less than in the glucose samples, whereas the serum loss of both were similar at the same sugar concentration. This was due to the MDX forming a continuous network in the whipped cream and blocking water, thus causing less serum loss (Figure 4B).

## 4. Conclusions

The influence of glucose and corn syrup on the physical characteristics of emulsions and the whipping properties of whipped creams have been investigated. Glucose and corn syrup were the main compositions of the whipped creams, which contributed greatly to the emulsion characteristics, partial coalescence, whipping properties and stability of whipped cream. In emulsions, a higher concentration of sugar decreased the droplet size and increased interfacial protein concentration and viscosity of the emulsion. In eventual whipped cream, there was a shorter *t_op_*, higher degree of fat coalescence, higher firmness and higher stability as sugar concentration increased. As a result, the addition of 25–30 wt% glucose or corn syrup could prevent the formation of large fat aggregation in emulsions and improve the whipping properties of whipped cream.

Compared with glucose, there were fewer numbers of irregularly shaped aggregates and higher stabilities of emulsion samples with corn syrup at the same sugar concentration, due to MDX. At 10–20 wt% sugar concentration, the whipped cream with corn syrup exhibited lower fat coalescence and interfacial protein concentration, which caused higher serum loss. However, at 25–30 wt% corn syrup concentration, the MDX network could block water proven by LF-NMR, causing lower serum loss even if there was a relatively lower fat coalescence degree and interfacial protein concentration.

Therefore, it could be deduced that the emulsions or creams would be well stabilized by a partial replacement of glucose by corn syrup or MDX. MDX possesses high potential as an alternative ingredient to produce low-sugar whipped cream in the future. In addition, our research could provide a reference for preparation and selection of sugars for whipped cream.

## Figures and Tables

**Figure 1 foods-11-01195-f001:**
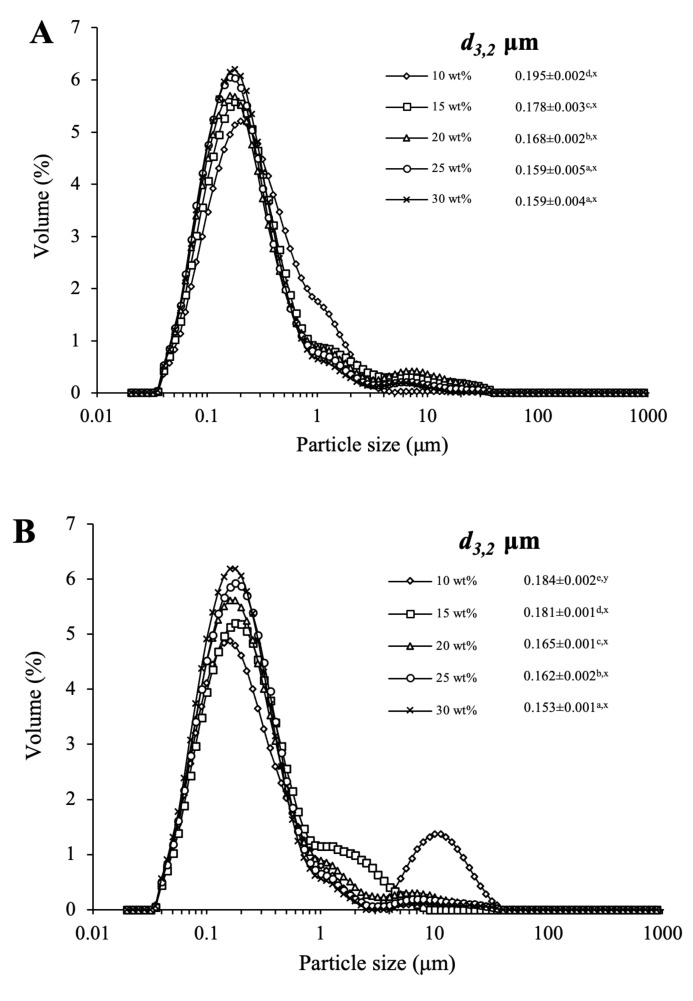
Particle size distribution and microstructure of the emulsions at different sugar concentrations. (**A**,**B**) represent particle size distribution of the emulsions with glucose (**A**) and corn syrup (**B**) before whipping. (**C**,**D**) represent microstructures of the emulsions with glucose (**C**) and corn syrup (**D**) before whipping, respectively. Different letters of a–e indicate significant differences (*p* < 0.05) among the various sugar levels. Different letters of x, y indicate significant difference (*p* < 0.05) between glucose and corn syrup in the same concentrations.

**Figure 2 foods-11-01195-f002:**
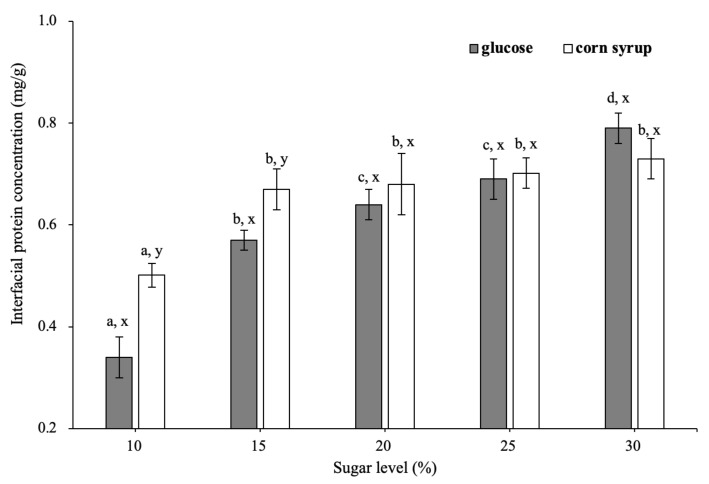
Interfacial protein concentration of the emulsions at different sugar concentrations. Different letters of a–d on the top of columns represent significant differences (*p* < 0.05) among the various sugar levels. Different letters of x, y on the top of columns represent significant difference (*p* < 0.05) between glucose and corn syrup at the same concentration.

**Figure 3 foods-11-01195-f003:**
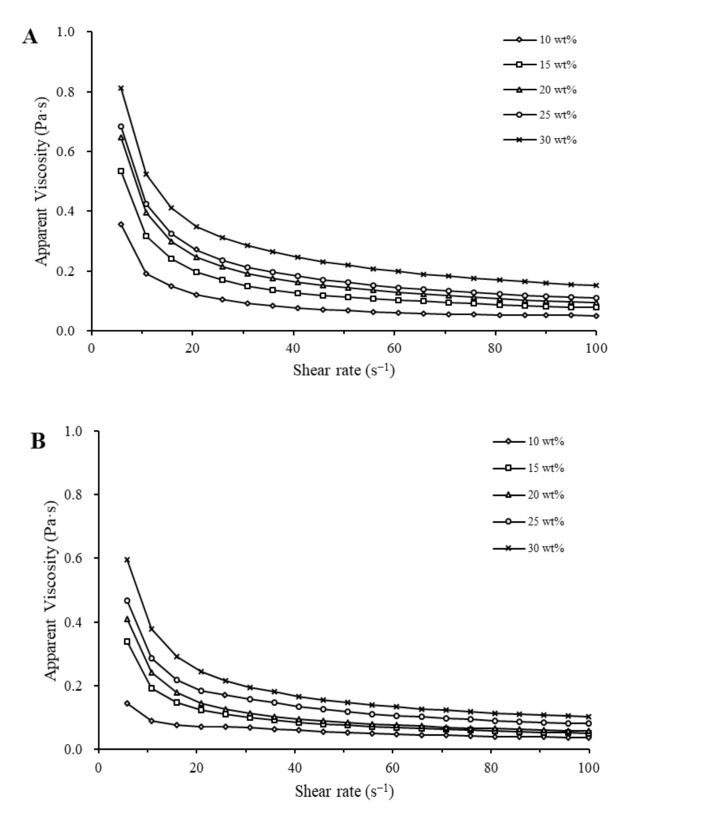
Interfacial protein concentration of the emulsions with glucose (**A**) and corn syrup (**B**) at different sugar concentrations.

**Figure 4 foods-11-01195-f004:**
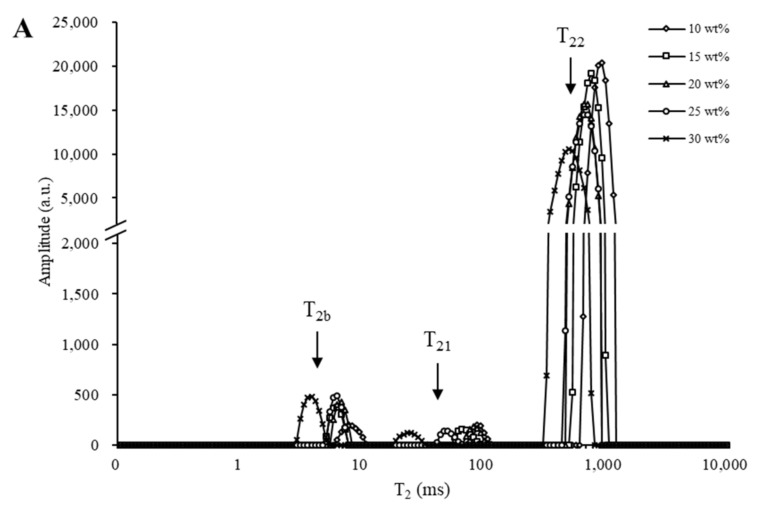
Distribution of T_2_ relaxation time of whipped creams with glucose (**A**) and corn syrup (**B**) at different sugar concentrations.

**Figure 5 foods-11-01195-f005:**
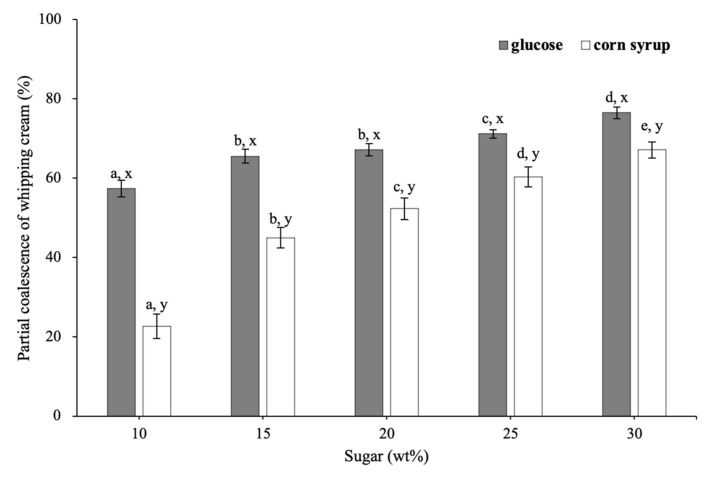
Partial coalescence of whipped creams at different sugar concentrations. Different letters of a–e on the top of columns represent significant differences (*p* < 0.05) among the various sugar levels. Different letters of x, y on the top of columns represent significant difference (*p* < 0.05) between glucose and corn syrup in the same concentration.

**Figure 6 foods-11-01195-f006:**
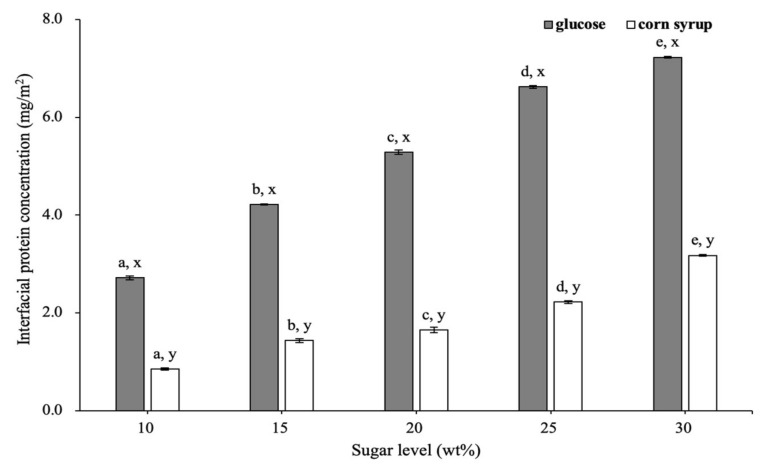
Interfacial protein concentration of whipped creams at different sugar concentrations. Different letters of a–e on the top of columns represent significant differences (*p* < 0.05) among various sugar levels. Different letters of x, y on the top of columns represent significant difference (*p* < 0.05) between glucose and corn syrup in the same concentration.

**Figure 7 foods-11-01195-f007:**
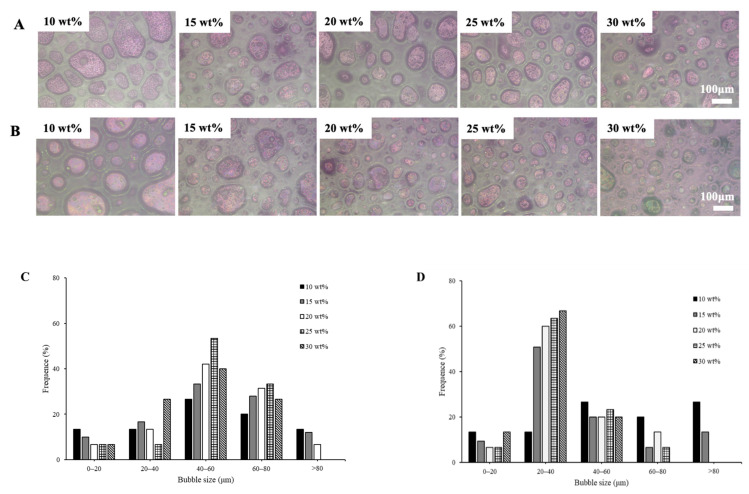
PLM images and the corresponding air bubble frequency distributions of whipped creams at different sugar concentrations. ((**A**,**B**) represent polarized optical microscopic images of whipped creams with glucose (**A**) and corn syrup (**B**). (**C**,**D**) represent the corresponding air bubble frequency distributions of whipped creams with glucose (**C**) and corn syrup (**D**)).

**Figure 8 foods-11-01195-f008:**
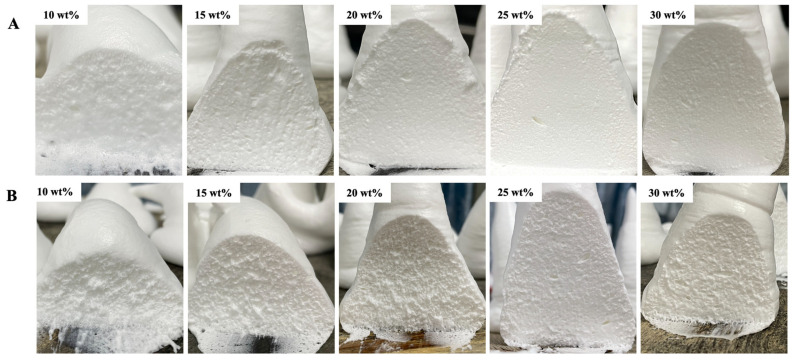
The images of cross-sections of whipped creams at different sugar concentrations. (**A**,**B**) represent the images of cross-sections of whipped creams with glucose (**A**) and corn syrup (**B**) after 6 h storage.

**Table 1 foods-11-01195-t001:** Whipping properties and serum loss of whipped creams with addition of different sugars at different concentrations.

Sample	Sugar Level (wt%)	*t_op_* (s)	Overrun (%)	Firmness (N)	Serum Loss (%)
Creams-glucose	10	980 ± 15 ^d,x^	264 ± 13 ^a,x^	1.25 ± 0.10 ^a,x^	26.09 ± 1.92 ^d,x^
	15	611 ± 9 ^c,x^	276 ± 10 ^a,x^	2.01 ± 0.09 ^b,x^	10.26 ± 0.51 ^c,x^
	20	447 ± 5 ^b,x^	335 ± 7 ^c,x^	2.61 ± 0.06 ^c,x^	6.31 ± 0.21 ^b,x^
	25	433 ± 5 ^b,x^	327 ± 10 ^b,c,x^	3.14 ± 0.03 ^d,x^	0.42 ± 0.15 ^a,x^
	30	390 ± 4 ^a^	306 ± 4 ^b,x^	3.82 ± 0.10 ^e,x^	0.23 ± 0.11 ^a,x^
Creams-corn syrup	10	1033 ± 17 ^e,y^	376 ± 4 ^e,y^	1.23 ± 0.03 ^a,x^	35.50 ± 1.13 ^d,y^
	15	838 ± 13 ^d,y^	330 ± 6 ^d,y^	2.02 ± 0.02 ^b,x^	20.81 ± 1.04 ^c,y^
	20	758 ± 10 ^c,y^	300 ± 6 ^c,y^	2.60 ± 0.09 ^c,x^	14.12 ± 0.87 ^b,y^
	25	711 ± 12 ^b,y^	261 ± 4 ^b,y^	3.06 ± 0.02 ^d,y^	0.54 ± 0.17 ^a,x^
	30	510 ± 14 ^a,y^	235 ± 9 ^a,y^	3.19 ± 0.03 ^d,y^	0.28 ± 0.13 ^a,x^

(Creams-glucose and Creams-corn syrup represent whipped creams with glucose and corn syrup, respectively. The *t_op_* represents the optimal whipping time. Different letters of a–e in the same column indicate significant differences (*p* < 0.05) among the various sugar levels. Different letters of x, y indicate significant difference (*p* < 0.05) between glucose and corn syrup in the same concentration.)

## Data Availability

Data is contained within the article and Appendix A.

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
