# Peer review of "Effects of Glucose and Corn Syrup on the Physical Characteristics and Whipping Properties of Vegetable-Fat Based Whipped Creams"

_foods, 2022, doi:10.3390/foods11091195_

Round 1
Reviewer 1 Report
The article is devoted to the study of the effect of glucose and corn syrup on the properties of emulsion systems and whipped cream with added sugar. Sodium caseinate and compound stabilizers were used as a stabilizer of studied systems. The problem considered in the work is certainly relevant from the point of view of creating emulsion culinary products and creams with a reduced sugar content. The paper considers the effect of sugar and maltodextrin on various physicochemical properties of emulsion systems - dispersion, structure, protein concentration in the stabilizing layer, viscosity. The properties of the cream were determined - water distribution, overrun, density, whey loss and microstructure. The authors used a set of modern instrumental methods, including light scattering, microscopy, rheological methods, low-field pulsed NMR, laser diffraction.
The article is structured clearly and logically, the conclusions are confirmed by experimental data.
Author Response
We sincerely thank you for your support and suggestions, which are a big encouragement for us.

Reviewer 2 Report
The manuscript entitled " Effects of glucose and corn syrup on the physical characteristics and whipping properties of non-dairy whipped creams" is an interesting topic that could be of interest for readers. However, there are some details to take into account.
INTRODUCTION
Line 59. Add the meaning of “high-DE” and “low-DE MDX” for better understanding.
Line 68-73. This paragraph fits better in the materials and methods section than introduction
RESULTS AND DISCUSSION
Line 213-219. Please, check the redaction of this paragraph.
Author Response
Responses for Reviewer 2
The manuscript entitled " Effects of glucose and corn syrup on the physical characteristics and whipping properties of non-dairy whipped creams" is an interesting topic that could be of interest to readers. However, there are some details to take into account.
We are very grateful to you for reviewing the paper so carefully. We have carefully considered your advice and made relevant revises; subsequently, these changes are marked up using the “Track Changes” function in revised manuscript.
1.Comment: Line 59. Add the meaning of "high-DE" and "low-DE MDX" for better understanding.
Reply: Thank your kind suggestions. We have added definitions and more details about DE in Line 250-255.
2.Comment: Line 68-73. This paragraph fits better in the materials and methods section than introduction
Reply: Thanks for the reviewer’s comment. We have recondensed and rewritten this paragraph and made it right in the new revised manuscript Line 269-276.
3.Comment: Line 213-219. Please, check the redaction of this paragraph
Reply: Thanks for the reviewer’s careful check. After carefully checking the results of this paragraph, we revised the paragraph and emphasized the importance of particle size distribution for emulsion stability.

Reviewer 3 Report
The abstract should be more informative by giving real results rather than elastic sentences. Important and main contents should be given. Support the results with some quantitative data. Moreover, no conclusions are provided.
The introduction is very incomplete and desperately needs to be rewritten and seriously revised.
The authors should mention the reason for selecting these sweeteners clearly and explain the objective of this study and its impact.
Line 80: The Compound stabilizers used must be clear about what compounds it contains. Because each of these components can be effective in the observed results. It is not at all acceptable to add an anonymous compound to a formula without first introducing it.
Line 81: When you have sodium caseinate in your product formulation, how do you describe your product as non-dairy? Is not sodium caseinate extracted from milk?
Results and discussion section: Descriptions of the observed phenomena in the results and discussion section are not sufficient and should be completed.
Table 1: I did not find a good explanation for the fact that in the first 2 concentrations of cream containing glucose, overrun was increased. Please state the reason for this observation with strong reasons.
Section 3.2.7: First, the size of the air bubbles has not been measured, which must be done. Second, not all bubbles are the same, and their distribution must be determined. Because in some shapes there are some large bubbles but the rest of the bubbles are smaller.
Conclusion: what is the future of your findings? The conclusion is not insightful, what are suggestions?
Author Response
Responses for Reviewer 3
Specific comments:
1.Comment: The abstract should be more informative by giving real results rather than elastic sentences. Important and main contents should be given. Support the results with some quantitative data. Moreover, no conclusions are provided.
Reply: Thank you very much for reminding kindly us to pay more attention to the emphasis of the Abstract, which means a lot to us. After carefully collecting and summarizing the results of our study, we have rewritten the Abstract. Moreover, we added quantitative data, real results and main conclusions, and modified the overall presentation of the Abstract, which was shown in the revised manuscript.
2.Comment: The introduction is very incomplete and desperately needs to be rewritten and seriously revised.
Reply: Thank your kind reminders. We are sorry for our incomplete and vague expression causing the description and understanding in “Introduction” part. Therefore, we have carefully organized the thoughts of our study and summarized them as follows:
(1) Firstly, in paragraphs 1-2 of the Introduction, the composition of whipped cream and the function of each component are introduced.
(2) Secondly, in paragraphs 3-4 of the Introduction, vegetable-fat based whipped cream has a huger advantage in cost and stability than butter-based whipped cream, but its high content of sugar does not match the needs of health. Moreover, few works focused on the mechanism involved in whipping properties of whipped cream formulated with different sugars.
(3) Then, in paragraphs 5-7 of the Introduction, analyzing the roles of sugar in a simple simulation system, ice cream and cake indicates that the effects of sugar in cream is multifaceted and complex.
(4) Finally, in paragraphs 8-9 of the Introduction, the ambiguous mechanism of sugar in emulsion-based foam system and the demand of the market, determines the purpose of our research.
3.Comment: The authors should mention the reason for selecting these sweeteners clearly and explain the objective of this study and its impact.
Reply: We gratefully appreciate your comments. According to your advice, we have rewritten paragraphs 4-7 of the introduction.
In paragraphs 5-7, we have rewritten this part to review the previous works and explain the functionality of sugars in aerated systems.
In paragraph 8, we summarize the inadequacy of previous work that the mechanisms related to the effect of sugar on whipping properties are unclear.
Reply: We gratefully appreciate your comments. 3 kinds of sugars (glucose, sucrose, and corn syrup) which are widely used in whipped cream have been used to prepare whipped creams and their effects on the qualities of the corresponding whipped cream have been investigated. Based on the result of preliminary experiments, there were similar whipping properties and qualities of whipped creams prepared with sucrose and glucose, and sucrose was seldomly added to whipped cream. Consequently, the glucose and corn syrup with greatly different structures were selected and the significant difference in their types and concentrations on the whipping properties and stability of whipped cream was observed in this study.
4.Comment: The Compound stabilizers used must be clear about what compounds it contains. Because each of these components can be effective in the observed results. It is not at all acceptable to add an anonymous compound to a formula without first introducing it.
Reply: Thanks for the reviewer’s comment. Compound stabilizers 8022, composed of polyglycerol ester of fatty acid, sodium stearyl lactate, sucrose ester S1170, xanthan gum, carrageenan, guar gum, sodium dihydrogen phosphate, and dibasic sodium phosphate, were provided by Guangdong Wenbang Biotechnology Co. (Zhaoqing, China). We have added the specific composition of compound stabilizers in Line 377-379.
5.Comment: Line 81: When you have sodium caseinate in your product formulation, how do you describe your product as non-dairy? Is not sodium caseinate extracted from milk?
Reply: Thank your kind reminder. The definition of non-dairy whipped cream is relatively vague in China, and usually refers to products prepared with vegetable oils or hydrogenated vegetable oils. To avoid confusion, “non-dairy whipped cream” was corrected as “vegetable-fat based whipped cream” (Kim et al., 2013).
Kim, H. J. , Bot, A. , ICMD Vries, Golding, M. , & Pelan, E. G. . (2013). Effects of emulsifiers on vegetable-fat based aerated emulsions with interfacial rheological contributions. Food Research International, 53(1), 342-351.
6.Comment: Results and discussion section: Descriptions of the observed phenomena in the results and discussion section are not sufficient and should be completed.
Reply: Thanks for the reviewer’s comments. We are sorry for our vague expression of observed phenomena causing the description and discussion rough at “Results and discussion” part. According to the reviewer’s comments, we have been reviewing this part carefully and doing the following revises to make the phenomena and structure clearer.
In Section 3.1.1, we revised and added quantitative data to make the phenomena completer in Line 843-845.
In Section 3.1.3, we added the phenomenon that the decreasing rate of apparent viscosity increased gradually as sugar concentration increased, then explained the reason in Line 983-985.
In Section 3.2.1, we added quantitative data to show the T2b of creams with glucose and corn syrup decreased with the increase in sugar concentration in Line 1016-1018, which make the structure completer.
In Section 3.2.3, we added the description to explain the phenomena that the interfacial protein concentration increased as sugar concentration increased in Line 1082-1084 and 1087-1089.
7.Comment: Table 1: I did not find a good explanation for the fact that in the first 2 concentrations of cream containing glucose, overrun was increased. Please state the reason for this observation with strong reasons.
Reply: Thank your kind suggestions. We added an explanation in Line 1157-1162. In brief, the whipping process involves the bubble being incorporated and broken. The weak structure formed during whipping process could entrap the air thus causing lower overrun. But, in corn syrup samples, the existence of MDX can retarding bubbles broken during the whipping process leading to higher overrun even at lower corn syrup concentrations.
8.Comment: Section 3.2.7: First, the size of the air bubbles has not been measured, which must be done. Second, not all bubbles are the same, and their distribution must be determined. Because in some shapes there are some large bubbles but the rest of the bubbles are smaller.
Reply: Thank the reviewer for pointing out our negligence in the figures’ discussion. According to your recommendation, air bubble size distribution was analyzed by using a procedure of ImageJ software, which showed in Fig.7C and 7D of the revised manuscript. Analyzing air bubble size distribution, it could be clearly observed that the size of air bubbles became more uniform as sugar concentration increased. Furthermore, at 25 and 30 wt% of sugar concentration, the average bubbles size of whipped creams with glucose was relatively bigger than that of corn syrup at the same sugar concentration. More explanation was supplied in Line 1191-1197 of the revised manuscript.
9.Comment: Conclusion: what is the future of your findings? The conclusion is not insightful, what are suggestions?
Reply: Thank your kind suggestions. We have listed the potential applications of maltodextrin as an alternative ingredient to prepare low-sugar whipped cream in the future in the Conclusion. Moreover, we have revised the conclusion to make it clearer. The suggestion was that our research would provide reference to preparation and selection of sugars for whipped cream.

Round 2
Reviewer 3 Report
It could be accepted.
Author Response
Thanks again for your support and suggestions, which are a big help and encouragement for us.